# Partially Hydrolysed Whey-Based Formulae with Reduced Protein Content Support Adequate Infant Growth and Are Well Tolerated: Results of a Randomised Controlled Trial in Healthy Term Infants

**DOI:** 10.3390/nu11071654

**Published:** 2019-07-19

**Authors:** Jacques Rigo, Stefanie Schoen, Marc Verghote, Bart van Overmeire, Wivinne Marion, Marieke Abrahamse-Berkeveld, Philippe Alliet

**Affiliations:** 1Department of Pediatrics, Neonatal Unit, University of Liege, CHU–CHR Citadelle, Boulevard du XIIème de Ligne 1, 4000 Liege, Belgium; 2Danone Nutricia Research, Uppsalalaan 12, 3584 CT Utrecht, The Netherlands; 3Department of Pediatrics, CHR Namur, Avenue Albert Premier 185, 5000 Namur, Belgium; 4Department of Neonatology, ULB Erasme, Lenniksebaan 808, 1070 Brussels, Belgium; 5Department of Neonatology, Clinique Saint Vincent, Rue François Lefèbre 207, 4000 Rocourt, Belgium; 6Department of Pediatrics, Jessa Hospital, Stadsomvaart 11, 3500 Hasselt, Belgium

**Keywords:** safety, protein, gastrointestinal tolerance, partially hydrolysed, infant formula, growth

## Abstract

The current study aimed to investigate growth, safety and tolerance of partially hydrolysed infant formulae in healthy full-term infants. Fully formula-fed infants were randomised ≤14 days of age to receive a partially hydrolysed whey formula with 2.27 g protein/100 kcal (pHF2.27) or the same formula with 1.8 g or 2.0 g protein/100 kcal (pHF1.8 and pHF2.0) until 4 months of age. The primary outcome was equivalence in daily weight gain within margins of ± 3 g/day; comparison with WHO Child Growth Standards; gastrointestinal tolerance parameters and number of (serious) adverse events were secondary outcomes. A total of 207 infants were randomised, and 61 (pHF1.8), 46 (pHF2.0) and 48 (pHF2.27) infants completed the study per protocol. Equivalence in daily weight gain was demonstrated for the comparison of pHF1.8 and pHF2.27, i.e., the estimated difference was −1.12 g/day (90% CI: [−2.72; 0.47]) but was inconclusive for the comparisons of pHF2.0 and pHF2.27 with a difference of −2.52 g/day (90% CI: [−4.23; −0.81]). All groups showed adequate infant growth in comparison with the World Health Organization (WHO) Child Growth Standards. To conclude, the evaluated partially hydrolysed formulae varying in protein content support adequate growth and are safe and well tolerated in healthy infants.

## 1. Introduction

Human milk represents the “gold standard” in infants’ nutrition due to its important nutritional and functional properties [1]. When newborn infants cannot receive human milk, they require specific infant milk formula produced according to stringent regulations and directives [2,3,4,5], and to provide nutritional and functional properties as close as possible to those of human milk.

It has been recommended that infants with a family history of allergy who are not (exclusively) breastfed are provided with a partially (pHF) or extensively hydrolysed formula (eHF) with confirmed reduced allergenicity [6]. Hydrolysed formulae were originally introduced as an elemental diet to treat intractable diarrhoea [7], to manage allergic diseases (eHF) or to prevent allergic sensitization (pHF) [8,9]. Nowadays, pHFs are also progressively used as random regular infant formulae in healthy infants who are not particularly at risk for allergy [10]. The hydrolysis of protein sources can significantly interfere with the bioavailability of nitrogen and other nutritional components of infant milk formula. In the 1990s, it was shown that the growth rate and protein efficiency could be impaired in infants fed various pHFs in comparison with infants fed either with human milk or with infant formula containing intact protein [11,12]. In addition, plasma amino acid concentrations measured in groups of the infants receiving different types of pHFs tended to reflect the amino acid composition of these pHFs with potential unfavourable imbalances in amino acids [13]. Hence in 2003, to compensate for these imbalances, the European Scientific Committee of Food report [2] fixed the minimal protein content for pHF at 2.25 g/100 kcal; a value higher than the recommended 1.8 g/100 kcal for infant formula containing intact protein.

Early growth differs between breastfed and formula-fed infants, which is possibly caused by the higher protein intake of formula-fed infants [14,15]. High protein intake is also associated with risks of obesity and chronic disease in later life [15,16,17]. Over the past decades, substantial efforts were made to optimise the composition of infant formula and reduce the protein content to more closely mimic the composition and potentially the functionality of human milk [18,19]. Similar investigations indicated that a reduced and balanced amino acid content of pHF might beneficially affect the infant’s nutritional status [20,21]. As a result, regulatory bodies reduced the minimum required protein content of pHF to 1.8/1.9 g/100kcal with the limitation that safety and suitability should be demonstrated [3,4,5,22].

The aim of the present study was to evaluate growth, safety and nutritional adequacy of pHFs with a protein content of 1.8, 2.0 and 2.27 g/100 kcal in healthy term infants during the first 4 months of life and to compare their growth to the World Health Organization (WHO) Child Growth Standards based upon growth of exclusively breastfed infants [23].

## 2. Materials and Methods

### 2.1. Participating Centres

This study was conducted in six study centres in Belgium (Centre Hospitalier Universitaire de Liège; Centre Hospitalier Régional de la Citadelle; Virga Jessa Ziekenhuis Hasselt; Centre Hospitalier Regional de Namur; Université libre de Bruxelles Hôpital Érasme; Clinique Saint Vincent Rocourt). All centres obtained approval of their independent local Ethical Review Board. The study was conducted in compliance with the Declaration of Helsinki and the International Conference on Harmonization guidelines for Good Clinical Practice and with the Belgium laws and regulations. The study was registered in the Dutch trial register (registration number NTR2128).

### 2.2. Subjects and Study Design

The study was designed as a prospective, randomised, controlled, double blind, equivalence study with three parallel groups. Exclusively formula-fed, healthy term infants, with a gestational age between ≥37 weeks and ≤42 weeks, postnatal age of ≤14 days, and birth weight of ≥2.5 kg and ≤4.5 kg were eligible for participation. Exclusion criteria were defined as conditions or illnesses that could interfere with the study, special dietary needs, or participation in any other study. The investigator obtained the informed consent from all parents/guardians (hereafter “parents”) and assessed the eligibility of the infant. Based on the order the infants were enrolled, the investigator assigned a randomisation number and opened a correspondingly numbered, opaque, sealed randomisation envelope, revealing the code of the study formula (A, B, C, D, E, F; two letters per study formula) that was assigned to the randomisation number beforehand. The randomisation sequence was generated using a disk operating system (DOS)-based program called RANDOM with sex and study site as strata by a statistician from Danone Nutricia Research who had no further involvement in the conduct of the study. Formulae were coded by the sponsor, and both the investigators and the infants’ parents were blind to the formulae.

### 2.3. Study Formulae

All formulae were powdered infant formula intended to provide complete nutritional support for infants in the first six months of life, manufactured in accordance with good manufacturing practices (ISO 22000) and compliant with Directive 2006/141/EC. The formulae were 100% partially hydrolysed whey protein-based and contained a specific prebiotic mixture, i.e., 90% short-chain galacto-oligosaccharides and 10% long-chain fructo-oligosaccharides (scGOS/lcFOS, 0.8 g/100 mL). The key difference between the experimental and control formulae was the protein content. The two experimental formulae contained either 1.8 or 2.0 g/100 kcal protein (pHF1.8 and pHF2.0), whereas the control formula contained 2.27 g/100 kcal protein (pHF2.27). The differences in protein content were paralleled by differences in carbohydrates (being 11.3, 11.1 and 10.8 g/100 kcal) to maintain an iso-caloric content of 66 kcal/100 mL of all formulae. All infants were to be fed ad libitum and exclusively with the formulae during the entire intervention period until 4 months of age.

### 2.4. Measurements

The primary outcome parameter was daily weight gain (g/day) from enrolment (baseline) until 4 months of age. Secondary growth outcome measures included length and head circumference. For the evaluation of safety and tolerance formula intake, gastrointestinal (GI) symptom parameters, adverse events and plasma parameters were recorded. After the baseline visit at ≤14 days of age, four additional visits were scheduled at 1, 2, 3 and 4 months of age. At 12 months of age, a post-intervention follow-up visit took place. Infant and parental characteristics were collected at the baseline visit. Growth outcome parameters were measured in duplicate at baseline and each visit thereafter. Infants were weighed naked, on calibrated electronic scales. Supine length of infants was measured using a standard measuring board. A non-stretchable measuring tape was used to measure head circumference. In case the measures deviated substantially (>100 g for weight and >5 mm for length and head circumference), a third measurement was obtained, and the measures closest together were averaged as outcome measurement. Weight-for-age *z*-score (WAZ), length-for-age *z*-score (LAZ), weight-for-length *z*-score (WLZ), and head circumference-for-age *z*-score (HCAZ) were calculated based on the WHO 2006 Child Growth Standards SAS Macro [24].

Parents received diaries to record study formula intake and GI symptoms daily during seven consecutive days preceding each visit. Severity of GI symptoms (vomiting, burping, flatulence, diarrhoea, constipation, diaper dermatitis, colic (cramps), regurgitation) was recorded once a day on a 4–point scale (absent, mild, moderate and severe). Investigators documented (serious) adverse events ((S)AEs) at each visit including onset, duration, severity and seriousness, relationship with the study formula, any actions that were taken, and the outcomes. (S)AEs were followed-up until they had abated or until a stable situation had been reached. At the end of the intervention, at 4 months of age, a blood sample was drawn from infants whose parents gave additional consent for this procedure. The nutritional status (albumin, pre-albumin), liver function (Alanine Aminotransferease (ALT), Aspartate Aminotransferase (AST)) and kidney function (creatinine, blood urea nitrogen (BUN)) were analysed at the local laboratory of each study site. Insulin-like growth factor–1 (IGF–1) was analysed centrally, at CHU-CHR Citadelle.

### 2.5. Statistics

Equivalence in daily weight gain (g/day) from baseline until 4 months of age between each experimental formula group (pHF1.8, pHF2.0) and the control group (pHF2.27) was to be confirmed. Equivalence of formula groups was demonstrated when the two-sided 90% confidence interval (CI) of the difference in estimated means of daily weight gain laid within the pre-defined equivalence margins of ± 3 g/day [25].

The required sample size for two one-sided statistical testing (using α = 0.05 and a power = 0.80) was 36 infants per formula group, according to the online tool used (www.sealedenvelope.com). Allowing for a drop-out rate of 30%, a total of 156 infants (52 per group) had to be enrolled. An interim analysis, including a re-estimation of the sample size was performed and led to an increase of the sample size to 207 infants (69 per formula group). Equivalence analyses for weight gain were performed using a linear regression model (GLM; General Linear Model), with the stratification factors sex and study site as a fixed effect. The impact of potential covariates (such as formula intake, birth weight and parental characteristics) on the estimate of the intervention effect were investigated in additional models. The secondary outcome parameters were analysed using the linear regression modelling approach to investigate differences between the formula groups.

For each formula group, the change in z-scores between baseline and 4 months, the z-scores at 4 months and lastly the *z*-scores at 12 months of age were evaluated post hoc by investigating the differences from the WHO Child Growth Standards [23] and by investigating equivalence with these standards. Formula groups were considered different from the WHO Child Growth Standards if the 95% CI of the estimated mean *z*-score of the change over time or of the estimated mean *z*-score at time points did not include zero, the median of the WHO Child Growth Standards. In addition, formula groups and the WHO Child Growth Standards were concluded equivalent if the calculated 90% CI of the estimated mean *z*-score of the change over time or of the estimated mean *z*-score at time points lay entirely within the predefined margin of ± 0.5 standard deviation (SD) of the WHO Child Growth Standards, a bandwidth considered to be indicative for adequate growth. For *z*-scores after baseline Parametric Curves models were applied, which described the development of growth parameters over time by a second-order polynomial curve and included study site and *z*-score at baseline as fixed effects. To facilitate the investigation and interpretation of the specific effect of the study formulas on growth outcomes, *z*-scores after baseline were estimated from the model assuming a 1:1 ratio of boys and girls, the largest study site as location and a baseline *z*-score of zero. Additionally, post-intervention *z*-scores at 12 months were analysed using an extended mixed model using the age variable as a categorical variable, including the same covariates.

To compare the effect of both experimental formulae with the control formula on GI symptoms, severity of each of the eight symptoms (vomiting, burping, flatulence, diarrhoea, constipation, diaper dermatitis, colic (cramps), regurgitation) was classified as 0 for absence of symptom, 1 for mild, 2 for moderate, and 3 for severe. For each infant, the mean score was calculated based on the diary information at each visit, if the diary was completed for at least 3 days. Infants’ study formula intake was compared between formula groups as mean intake (mL/day) per visit as well as the intake corrected for body weight (mL/kg/day) measured at visits. Differences in GI symptoms, study formula intake and blood parameters were evaluated using the non-parametric Van Elteren test (VE) correcting for sex.

The Per Protocol (PP) population was used for the analysis of the primary outcome parameter, as well as for the post hoc analyses comparing the formula groups with the WHO Child Growth Standards. The Intention-To-Treat (ITT) population was used for the analyses of the GI symptoms. Adverse events were analysed for the All-Subjects-Treated population, which did not differ from the ITT population.

All statistical analyses were performed using SAS^®^ for Windows (SAS Enterprise Guide 4.3 or higher, SAS Institute Inc., Cary, NC, USA).

## 3. Results

### 3.1. Subject Characteristics

From January 2010 to September 2011, 207 infants were screened for eligibility and randomised to one of the three study formulae. One year later, in August 2012, the last subject completed the study (Figure 1).

One hundred sixty-two infants completed the intervention period with 63 infants in pHF1.8, 47 infants in pHF2.0 and 52 infants in pHF2.27. The overall drop-out rate was 22% (13% in pHF1.8, 30% in pHF2.0, 24% in pHF2.27). The drop-out rate, as well as reasons for early termination, were not statistically significantly different between the experimental and the control formula group. The main reason for early termination was the occurrence of an (serious) adverse event ((S)AE) (seven infants in pHF1.8; 17 infants in pHF2.0; 11 infants in pHF2.27), other reasons were lost-to-follow-up (two infants in pHF1.8; one infant in pHF2.27), non-specified reason (two infants in pHF2.0; two infants in pHF2.27), withdrawal of informed consent (one infant in pHF2.0; one infant in pHF2.27) or protocol violation (one infant in pHF2.27).

Apart from small differences in the ratio of boys and girls, other infant characteristics at birth (Table 1) and infants’ anthropometric data at baseline (Appendix A) were not apparently different between the formula groups for the PP as well as ITT population (data not shown). In addition, no apparent differences in the parental characteristics between the formula groups were observed (Appendix A).

### 3.2. Formula Intake

The average formula intake during the intervention period was similar in the formula groups. No statistically significant differences between the formula groups at any time point were identified when correcting the formula intake for body weight (Appendix A).

### 3.3. Growth Outcomes

#### 3.3.1. Growth Outcomes—Comparison Between Formula Groups

Equivalence in daily weight gain (g/day) was demonstrated for the comparison of pHF1.8 and pHF2.27 (Figure 2). The difference in estimated means (90% CI) between the groups was −1.1 g/day (−2.7; 0.5), with an observed mean (SD) daily weight gain of 29.6 (5.8) g/day in pHF1.8 and 31.0 (5.6) g/day in pHF2.27. Equivalence in daily weight gain was inconclusive for the comparison of pHF2.0 and pHF2.27, as the lower CI crossed the lower equivalence margin of −3 g/day (Figure 2). The difference in estimated means was −2.5 g/day (−4.2; −0.8); the observed daily weight gain for pHF2.0 was 28.0 (4.4) g/day.

The impact of potential covariates (such as formula intake, birth weight and parental characteristics) were investigated, but they did not explain the differences between the groups (data not shown).

No statistically significant differences were observed between the formula groups in length and head circumference gain during the intervention period.

#### 3.3.2. Growth Outcomes—Comparison with the WHO Child Growth Standards

Apart from the evaluation of growth outcomes between the study formula groups, the growth outcomes of each formula group were compared with the WHO Child Growth Standards. We evaluated the growth outcomes using statistical tests for differences and equivalence in comparison with the WHO Child Growth Standards. This allows to assess proximity to the WHO Child Growth Standards (statistically different from zero, the median of the WHO Child Growth Standards) and to verify via an equivalence analysis if any potentially observed differences are clinically relevant.

First, the change in *z*-scores from baseline to 4 months was investigated. In pHF1.8 and pHF2.27, the change in WAZ was not statistically significantly different from the WHO Child Growth Standards. In contrast, in pHF2.0, the WAZ decreased statistically significantly over time (Table 2, column A). The LAZ increased in all formula groups over time, and the increase was statistically significantly higher in pHF2.0 and pHF2.27 in comparison with the WHO Child Growth Standards. In all formula groups, the change in WLZ was not statistically significantly different from the WHO Child Growth Standards. The increase in HCAZ was statistically significantly higher in pHF1.8 and pHF2.27 in comparison with the WHO Child Growth Standards but not in pHF2.0.

In pHF1.8 and pHF2.0, the change in *z*-scores of all growth parameters was equivalent to the WHO Child Growth Standards. In pHF2.27 the change in WAZ (Figure 3) and WLZ was equivalent, the change in LAZ and HCAZ was inconclusive as the CI crossed the upper equivalence margins of + 0.5 SD (Table 2 column (B)).

At the end of the intervention, at the infants’ age of 4 months, the *z*-scores of all growth outcome parameter of each formula group were relatively close to the median (0) of the WHO Child Growth Standards. Apart from a higher LAZ in pHF1.8 and pHF2.0 and a higher HCAZ in pHF1.8, no statistically significant differences were observed in growth outcome parameters between the two experimental formula groups and the WHO Child Growth Standards (Table 3 column A, Figure 4). In pHF2.27, except for the WLZ, all other growth parameters were statistically significantly higher than the WHO Child Growth Standards (Table 3 column A, Figure 4).

In pHF1.8 and pHF2.0 *z*-scores of all growth parameters at 4 months of age were equivalent to the WHO Child Growth Standards, whereas in pHF2.27, only the WAZ was equivalent. The equivalence analyses for the other *z*-scores in pHF2.27 were inconclusive as the CI crossed the upper equivalence margins of + 0.5 SD (Table 3 column B).

At 12 months of age, *z*-scores of all growth outcome parameters were somewhat but statistically significantly higher than the WHO Child Growth Standards in pHF1.8 and pHF2.27, whereas in pHF2.0 only the HCAZ was statistically significantly higher (Table 4, column A).

In the pHF1.8 group, all *z*-scores were equivalent, except the HCAZ, which was inconclusive as the 90% CI crossed the upper equivalence margin of + 0.5 SD. In pHF2.0, equivalence to WHO Child Growth Standards was demonstrated for all *z*-scores. In pHF2.27, equivalence was only demonstrated for LAZ, but inconclusive for all the other *z*-scores of growth outcomes as the upper margins of + 0.5 SD were crossed (Table 4, column B).

### 3.4. Blood Parameter

At 4 months of age, 111 blood samples were collected and analysed (43 samples in pHF1.8, 33 samples in pHF2.0 and 35 samples in pHF2.27). In both experimental formulae groups, the blood urea nitrogen (BUN) was statistically significantly lower than in pHF2.27 (*p* < 0.001, VE). In pHF1.8 the median (min; max) was 15.0 (5.1; 21.0) mg/dL, in pHF2.0 15.5 (5.6; 22.0) mg/dL, and in pHF2.27 20.5 (7.0; 28.0) mg/dL. No statistically significant differences were observed between the formula groups in the concentration of Insulin-like growth factor–1 (IGF–1). In pHF1.8 the median (min; max) was 106.5 (59.0; 231.9) ng/mL, in pHF2.0 97.6 (62.0; 176.0) ng/mL and in pHF2.27 101.5 (71.0; 203.5) ng/mL. Furthermore, no statistical differences in the nutritional status (albumin, pre-albumin), liver function (ALT, AST) and kidney function (creatinine) parameter were observed.

### 3.5. Parent-Reported Gastrointestinal Tolerance

The severity scores for vomiting, diarrhoea, constipation and diaper dermatitis were very low, i.e., close to absent (median severity was 0), at all time points for all formula groups (Appendix A). Nevertheless, the comparison between pHF1.8 and pHF2.27 revealed statistically significant differences, explained by lower maximum values in pHF1.8, for the severity of diarrhoea at 3 months (*p* = 0.027, VE), and for diaper dermatitis at 4 months (*p* = 0.020, VE). No statistically significant differences between the formula groups were observed in any of the other GI symptom parameters at any time point. Burping, flatulence, and colic (cramps) were mild at the age of 4 weeks (median approximately 1) in all formula groups, and the severity of these symptoms was even lower at the later visits. Regurgitation was mild in all formula groups at the age of 1 month which remained until the later visits.

### 3.6. Adverse Events

During the intervention period, 15 serious adverse events (SAEs) were reported for 15 infants (7.2% of 207 infants). The percentage of infants with one or more SAEs overall as well as by severity score was not statistically significantly different between formula groups, as the types were diverse and distributed over the body systems and the formula groups. There were in pHF1.8 six events in six subjects (8.3%) (preferred terms: fever, apnoea, atelectasis, pneumonia), in pHF2.0 four events in four subjects (6.0%) (preferred terms: meningitis, infection viral, pharyngitis, pneumonia) and in pHF2.27 five events in five subjects (7.4%) (preferred terms: gastroesophageal reflux, infection viral, pneumonia, surgical intervention, cerebral haemorrhage). The most frequently reported SAE was pneumonia (three events, three subjects in pHF1.8 and one event in one subject in pHF2.0 and pHF2.27. No statistically significant differences were observed in the percentage of infants with adverse events (AEs). The results also did not show any statistical differences between the formula groups in number of infants who experienced an AE, categorised in any of the body systems except for Ocular Disorders, term “conjunctivitis” (all AEs were mild in severity and causality was not related). There were 12 events in 10 infants (14.9%) in pHF2.0, while in pHF2.27 there were only two events in two infants (2.9%; *p* = 0.017 FE). Based on the evaluation of the occurrence of (S)AE, there was no safety concern raised during the study.

## 4. Discussion

This prospective, randomised, controlled, double blind equivalence study evaluated the nutritional adequacy of formulae containing partially hydrolysed whey protein with a reduced protein content of –10% (2.0 g/100 kcal) and –20% (1.8 g/100 kcal), in comparison with a currently marketed formula containing 2.27 g/100 kcal protein. We observed an equivalent daily weight gain for the pH1.8, but not for the pH2.0 group in comparison with the pH2.27 control group. The comparison with the WHO Child Growth Standards, based on growth of exclusively breastfed infants, confirmed equivalence with the WHO Child Growth Standards in change in WAZ from baseline until 4 months and in WAZ at 4 months for all formula groups including the pHF2.0 group. No safety concerns were raised based on the absence of clinically relevant (serious) adverse events. Hence, the current study demonstrated that both experimental formulae with reduced protein content supported adequate growth, were well tolerated and are safe for infants.

Our findings are in line with previous studies from Ahrens et al. [26] and Ziegler et al. [27], indicating adequate growth in infants fed whey-based pHF with reduced protein content (1.9 g/100 kcal) in comparison with a control formula containing either 2.3 g/100 kcal protein [26] or 2.4 g/100 kcal [27]. Ahrens et al. reported for the period between 28 and 112 days of age a median (IQR) daily weight gain of 28.5 (8.5) g/day for a formula without synbiotics and 28.9 (6.9) g/day for a formula with synbiotics. We obtained a mean (SD) daily weight gain of 27.4 (5.9) g/day in pH1.8, 26.0 (4.7) g/day in pH2.0 and 28.6 (5.8) g/day in pH2.27 for the same period. Similarly, Ziegler et al. reported for the period between 8 and 112 days of age a mean (SD) daily weight gain of 32 (4.7) g/day in boys and 28.2 (5.8) g/day in girls of the reduced protein formula group. In our study, for the period between baseline (8 days) until 4 months (112 days), the mean (SD) daily weight gain for the pH1.8 group was 32.1 (6.0) g/day in boys and 26.8 (4.0) g/day in girls, for the pH2.0 group: 28.7 (3.4) g/day in boys and 27.4 (5.1) g/day in girls and for the pH2.27 group: 32.8 (5.5) g/day in boys and 28.4 (4.6) g/day in girls.

Our growth outcomes of infants fed the pHF are also similar to those of infants fed reduced intact protein formulae as investigated by Alexander et al. [28] and Beghin et al. [29]. Alexander et al. [28] performed a pooled data analysis of 11 randomised controlled trials in healthy term infants fed whey-predominant formulae with a content of 1.8 g/100 kcal intact protein with or without additional ingredients (probiotics, prebiotics, or both). They reported WAZ estimates (95% CI) at 4 months of age for the formula groups without additional ingredients of −0.03 (−0.12, 0.05) and for the formula groups with additional ingredients of 0.14 (0.02, 0.26). We reported estimated mean (95% CI) WAZ for pHF1.8 of 0.01 (−0.19, 0.21) and for pHF2.0 of −0.03 (−0.22, 0.15).

Beghin et al. [29] reported a large randomised controlled trial evaluating growth in term infants fed from ≤1 week to 4 months of age one of three identical whey-predominant formula containing 2.1 g/100 kcal protein but differing by their SN2–palmitate content. The mean (SD) weight gain from inclusion to 4 months across the three groups was 27.8 (5.7) g/day (*n* = 488), which is slightly lower to our observed data in the reduced protein groups: pHF1.8: 29.6 (5.8) and pHF2.0: 28.0 (4.4) g/day, and considerably lower than that observed in our pHF2.27 group with 31.0 (5.6) g/day. In summary, in contrast to older studies suggesting that the nutritional value of formulae containing hydrolysed protein could be lower than that containing intact protein [11], the growth parameters in our study of the two experimental and the control formulae are comparable to the reports of recent studies, evaluating growth in healthy term infants fed intact whey dominant formula with a protein content of 1.8 or 2.1 g/100 kcal [28,29].

The composition of the three formulae in our study was identical except for the protein content of 1.8, 2.0 and 2.27 g/100 kcal and its compensation by carbohydrate content to maintain an iso-caloric value of 66 kcal/100 mL. This allowed the evaluation of the specific effect of a minimal change in protein content on growth in healthy term infants. In line with previous findings of Turck [30], the lower protein levels of 1.8 and 2.0 g/100 kcal did not result in a compensatory increase of formula intake. The similar volume intake in the formula groups indicated indirectly that the intended differences in protein intake were achieved, which were confirmed by the lower BUN values in both experimental groups compared with pHF2.27. Interestingly, the observed differences in total protein intake do not seem to drive growth outcomes proportionally, as the mean weight gain during the study was lowest in pHF2.0, whereas pHF1.8 was intermediate. Potentially, other factors influenced (more) significantly our growth outcome parameters, such as study population differences, reconstitution and feeding practices of the parents affecting the density of the formula and volume intakes [31], which might have been insufficiently captured via diaries.

Our study lacks a breastfed reference group, and we, therefore, evaluated growth outcomes using the WHO Child Growth Standards representing the growth of healthy term breastfed infants [23]. However, it should be noted that statistical methods for the comparison with the WHO Child Growth Standards seem to be controversial. In our study, we evaluated a difference from the median (*z*-score of 0) of the WHO Child Growth Standards, applying mixed models and evaluating estimated means with their 95% CIs of each study group. Additionally, we evaluated the equivalence to WHO Child Growth Standards to assess the clinical relevance of observed differences by using the estimated means with their 90% CI for simultaneous comparison with the upper and lower equivalence margin set at ± 0.5 SD, which we defined as clinically relevant margins. By contrast, others present only descriptive values (raw means), use non-inferiority evaluation with a margin of −0.5 SD and a 95% CI [32] or testing for differences with a 95% CI in combination with assessment of clinical relevance by using a margin of ± 0.5 SD [28]. Evident from the comparison between Table 3 and Appendix A, the application of mixed models correcting for potential imbalances between groups to estimate the intervention effects may result in a different interpretation of growth outcome parameters. The different statistical approaches used for evaluation of growth outcomes are likely to impair the comparison between studies and justify the need for statistical guidelines.

Although well established and providing relevant context for interpretation of growth outcomes of formula-fed infants when a breastfed reference group is lacking, the comparison with the WHO Child Growth Standards has some limitations. Several studies indicated that growth outcomes of breastfed infants do not follow these standards strictly. For example, several of the individual studies included in the pooled data-analysis of Alexander [28] investigated growth of breastfed infants and resulted in estimated mean *z*-scores of weight, length, BMI and head circumference which even lay outside the ± 0.5 SD of WHO Child Growth Standards at 4 months of age. Even more pronounced are differences between breastfed infants and the WHO Child Growth Standards during the first month of age [32,33]. The review of the WHO Child Growth Standards for weight shows a limited weight loss during the early days of life accounting, respectively, to about 1% of body weight after 1 day and a mean weight gain of 3% to 4% after 1 week. That contrasts to the early weight loss normograms recently published [33], which suggest that in breastfed infants the weight loss could reach up to 6% to 8% of the birth weight between 2 and 4 days of life with a return to birth weight between 6 to 10 days of life. Such a difference could explain the early decrease of WAZ after birth as frequently observed in growth studies in breastfed and formula-fed infants. For example, Spalinger et al. reported a decrease of WAZ from 0.04 at birth to −0.31 at 9 days of age [32]. This implies that caution is needed for the comparison of absolute (g/day) and relative (*z*-scores) weight gain values between growth studies depending on differences in their start age and the duration of the intervention.

In our study, the pHF2.27 formula group showed a limited but statistically significant higher WAZ than the WHO Child Growth Standards median at the end of the intervention at 4 months of age. This was paralleled by a higher LAZ and HCAZ, but not WLZ, indicative for proportional infant growth. Interestingly, the estimated mean WAZ at 4 months as well as its change between baseline and 4 months were equivalent to the WHO Child Growth Standard. Hence, although the growth of the infants in pHF2.27 is statistically different from the median WHO Child Growth Standards, we consider this growth to be adequate since it lays within the pre-defined equivalence margins defined as clinically relevant. The higher *z*-scores in pHF2.27 remain over time, although potentially influenced by other (dietary) factors during the post-intervention period and resulted in significantly higher *z*-scores than the WHO Child Growth Standard at 12 months, for which also the equivalence results remained inconclusive as the CIs crossed the upper equivalence margins. Up to now, the longer-term health effects of minimal differences in protein intakes as evaluated in our study is unknown as suggested by recent review or meta-analyses [34,35]. However, these observations together with the observed higher plasma BUN levels, do suggest that there is a window of opportunity to lower the protein/energy ratio for pHF to the same ranges than that recommended for the intact protein formulae.

## 5. Conclusions

In conclusion, partially hydrolysed protein formulae with a standard (2.27 g/100 kcal) or reduced protein content (1.8 or 2.0 g/100 kcal) support adequate growth, are well tolerated and safe for use in infants. Future longitudinal clinical studies are required to evaluate the potential long-term health effects of minimal differences in protein intakes during infancy.

## Figures and Tables

**Figure 1 nutrients-11-01654-f001:**
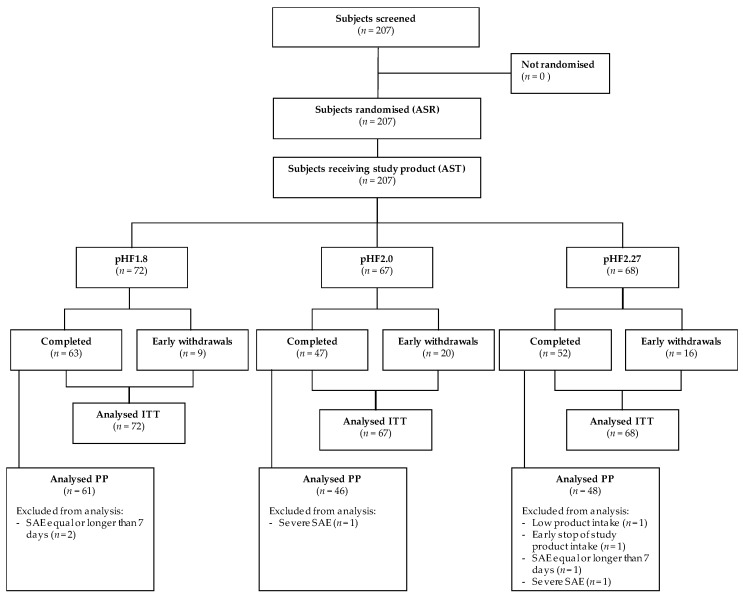
Flow Diagram. pHF, partially hydrolysed formula; PP, Per Protocol; ITT, Intention-To-Treat.

**Figure 2 nutrients-11-01654-f002:**
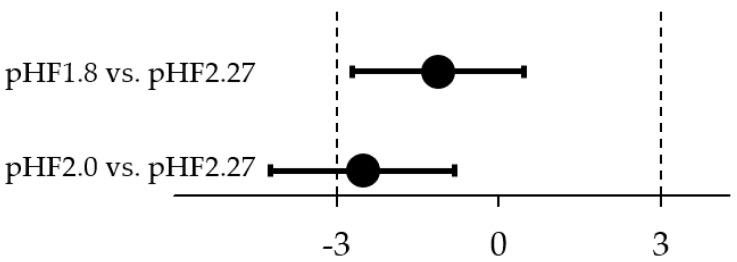
Graphical display of the difference in estimated means (90% CI) of weight gain per day between the two experimental formulae and the control formula group. The equivalence analysis was performed applying predefined margins of ± 3 g/day, using a linear regression model with the stratification factors sex and study site as a fixed effect.

**Figure 3 nutrients-11-01654-f003:**
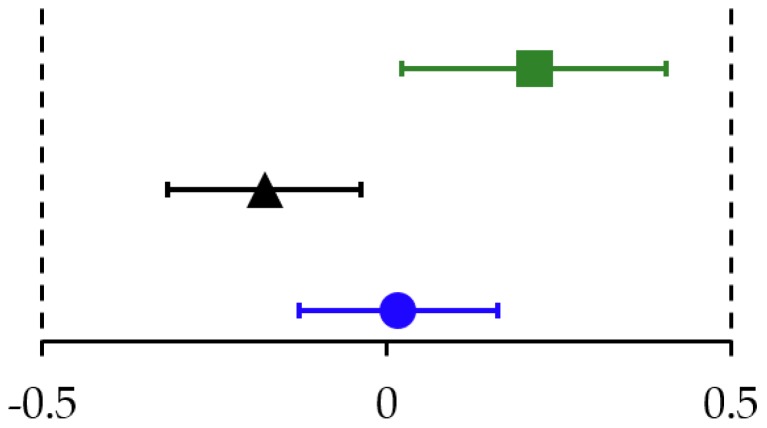
Estimated mean (90% CI) of change in weight-for-age *z*-scores from baseline to 4 months for pHF1.8 (blue), pHF2.0 (black) and pHF2.27 (green). Assessment of equivalence with the WHO Child Growth Standards, applying predefined margins of ± 0.5 SD.

**Figure 4 nutrients-11-01654-f004:**
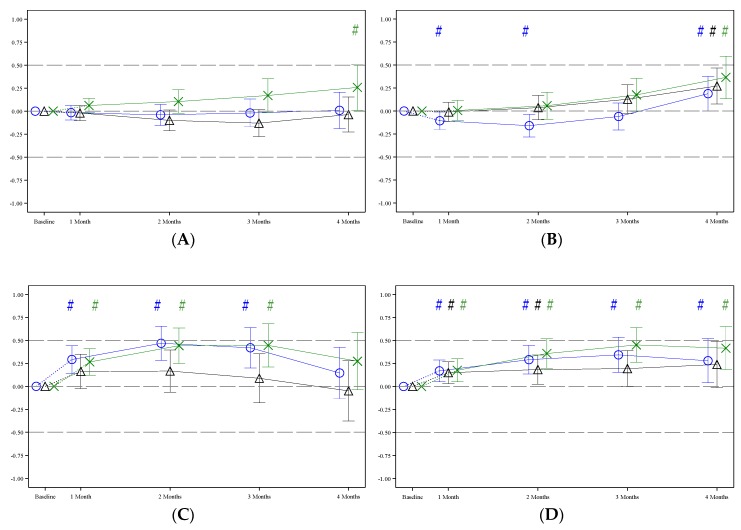
Estimated mean (95% CI) *z*-scores during the intervention period for pHF1.8 (blue), pHF2.0 (black) and pHF2.27 (green): (**A**) weight-for-age *z*-score; (**B**) length-for-age *z*-score; (**C**) weight-for-length *z*-score; (**D**) head circumference-for-age *z*-score. Post baseline estimates are derived for the largest study site, 1:1 ratio for boys and girls and a respective baseline *z*-score value of zero as fixed effects. For completeness, the observed (raw) *z*-scores are included in Appendix A. # = Statistically significant difference between a formula group and WHO Child Growth Standards.

**Table 1 nutrients-11-01654-t001:** Infant characteristics at birth of the Per Protocol (PP) population.

		pHF1.8 (*n* = 61)	pHF2.0 (*n* = 46)	pHF2.27 (*n* = 48)	Total (*n* = 155)
Sex					
male	*n* (%)	32 (52.5%)	22 (47.8%)	28 (58.3%)	82 (52.9%)
female	*n* (%)	29 (47.5%)	24 (52.2%)	20 (41.7%)	73 (47.1%)
Gestational age (weeks)	mean (SD)	39.3 (1.1)	39.5 (0.8)	39.3 (1.3)	39.3 (1.1)
Birth Weight (g)	mean (SD)	3337 (430)	3347 (420)	3332 (453)	3339 (432)
Length birth (cm)	mean (SD)	49.5 (1.9)	50.1 (2.1)	49.5 (2.2)	49.7 (2.1)
Birth Head Circumference (cm)	mean (SD)	34.2 (1.4)	34.4 (1.6)	34.1 (1.2)	34.2 (1.4)

SD, standard deviation. pHF, partially hydrolysed formula.

**Table 2 nutrients-11-01654-t002:** Estimated means of change in *z*-score of growth outcome parameters from baseline to 4 months of the PP population for (A) differences from WHO Child Growth Standards based on 95% CI and (B) equivalence with WHO Child Growth Standards based on 90% CI.

Parameter (*z*-Scores)	Group	Estimate	(A)	(B)
			Investigating a Difference from WHO Child Growth Standards	Investigating Equivalence with WHO Child Growth Standards, Considering a Margin of ± 0.5 SD
			(Difference from the Median (0))	
			95% Confidence Interval	90% Confidence Interval
Weight-for-age (WAZ)	pHF1.8	0.02	−0.16; 0.19	−0.13; 0.16
	pHF2.0	−0.18	−0.35; −0.01 #	−0.32; −0.04
	pHF2.27	0.21	−0.02; 0.44	0.02; 0.41
Length-for-age (LAZ)	pHF1.8	0.17	−0.003; 0.34	0.03; 0.31
	pHF2.0	0.28	0.12; 0.44 #	0.14; 0.42
	pHF2.27	0.35	0.16; 0.54 #	0.19; 0.51 *
Weight-for-length (WLZ)	pHF1.8	0.12	−0.13; 0.37	−0.09; 0.33
	pHF2.0	−0.17	−0.45; 0.11	−0.41; 0.06
	pHF2.27	0.24	−0.05; 0.54	−0.01; 0.49
Head circumference-for-age (HCAZ)	pHF1.8	0.29	0.10; 0.47 #	0.13; 0.44
	pHF2.0	0.07	−0.13; 0.27	−0.10; 0.24
	pHF2.27	0.47	0.29; 0.64 #	0.32; 0.61 *

# 95% CI = difference was confirmed between the formula group and WHO Child Growth Standards; the CI did not include 0. * 90% CI = equivalence was not confirmed between the study group and WHO Child Growth Standards, the CI included ± 0.5 SD. The model included study site, sex and baseline. Estimates are derived for the largest study site, 1:1 ratio for boys and girls and a respective baseline *z*-score value of zero as fixed effects.

**Table 3 nutrients-11-01654-t003:** Estimated *z*-scores of growth outcome parameters at 4 months age of the PP population, evaluating (A) differences from WHO Child Growth Standards based on 95% CI or (B) equivalence to WHO Child Growth Standards based on 90% CI.

Parameter (*z*-Scores)	Group	Estimate	(A)	(B)
			Investigating a Difference from WHO Child Growth Standards	Investigating Equivalence to WHO Child Growth Standards, Considering a Margin of ± 0.5 SD
			(Difference From the Median (0))	
			95% Confidence Interval	90% Confidence Interval
Weight-for-age (WAZ)	pHF1.8	0.01	−0.19; 0.21	−0.16; 0.18
	pHF2.0	−0.04	−0.23; 0.15	−0.20; 0.12
	pHF2.27	0.26	0.01; 0.50 #	0.05; 0.46
Length-for-age (LAZ)	pHF1.8	0.19	0.00; 0.38 #	0.03; 0.35
	pHF2.0	0.27	0.08; 0.47 #	0.11; 0.44
	pHF2.27	0.37	0.14; 0.60 #	0.17; 0.56 *
Weight-for-length (WLZ)	pHF1.8	0.15	−0.13; 0.43	−0.09; 0.38
	pHF2.0	−0.05	−0.38; 0.28	−0.32; 0.23
	pHF2.27	0.27	−0.04; 0.59	0.01; 0.54 *
Head circumference-for-age (HCAZ)	pHF1.8	0.28	0.04; 0.52 #	0.08; 0.48
	pHF2.0	0.24	−0.01; 0.49	0.03; 0.45
	pHF2.27	0.42	0.18; 0.65 #	0.22; 0.61 *

# 95% CI = difference was confirmed between the formula group and WHO Child Growth Standards; the CI did not include 0. * 90% CI = equivalence was not confirmed between the study group and WHO Child Growth Standards; the CI crossed the ± 0.5 SD margins. The model included study site, sex and baseline. Estimates are derived for the largest study site, 1:1 ratio for boys and girls and a respective baseline *z*-score value of zero as fixed effects.

**Table 4 nutrients-11-01654-t004:** Estimated (LS mean, 95% CI and 90% CI) *z*-scores of growth outcome parameters at 12 months age of the per protocol population to evaluate (A) differences from WHO Child Growth Standards using the 95% CI and (B) equivalence to WHO Child Growth Standards using the 90% CI.

Parameter (*z*-Scores)	Group	Estimate	(A)	(B)
			Investigating a Difference from WHO Child Growth Standards	Investigating Equivalence with WHO Child Growth Standards, Considering a Margin of ± 0.5 SD
			(Difference from the Median (0))	
			95% Confidence Interval	90% Confidence Interval
Weight-for-age (WAZ)	pHF1.8	0.21	0.03; 0.40 #	0.06; 0.37
	pHF2.0	0.06	−0.13; 0.24	−0.10; 0.2
	pHF2.27	0.35	0.12; 0.58 #	0.16; 0.54 *
Length-for-age (LAZ)	pHF1.8	0.20	0.02; 0.39 #	0.05; 0.36
	pHF2.0	0.16	−0.03; 0.35	0.00; 0.32
	pHF2.27	0.25	0.03; 0.48 #	0.06; 0.44
Weight-for-length (WLZ)	pHF1.8	0.29	0.05; 0.53 #	0.09; 0.49
	pHF2.0	0.20	−0.08; 0.48	−0.03; 0.44
	pHF2.27	0.40	0.15; 0.66 #	0.19; 0.62 *
Head circumference-for-age (HCAZ)	pHF1.8	0.51	0.29; 0.73 #	0.32; 0.69 *
	pHF2.0	0.25	0.01; 0.49 #	0.05; 0.45
	pHF2.27	0.59	0.40; 0.78 #	0.43; 0.75 *

# 95% CI = difference was confirmed between the formula group and WHO Child Growth Standards; the CI did not include 0. * 90% CI = equivalence was not confirmed between the study group and WHO Child Growth Standards, the CI included ± 0.5 SD. The model included study site, sex and baseline. Estimates are derived for the largest study site, 1:1 ratio for boys and girls and a respective baseline *z*-score value of zero as fixed effects.

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
