# Peer review of "Partially Hydrolysed Whey-Based Formulae with Reduced Protein Content Support Adequate Infant Growth and Are Well Tolerated: Results of a Randomised Controlled Trial in Healthy Term Infants"

_nutrients, 2019, doi:10.3390/nu11071654_

Round 1

Reviewer 1 Report

Authors have studied infants (growth) consuming different amounts of protein in the partially hydrolyzed whey-based formula.  The manuscript is interesting and authors have stated longitudinal studies are needed to fully understand the different amounts of protein intake impact on infants’ growth and development. Conclusions are reasonable and limitation are discussed. The following edits will strengthen the manuscript.

Comments:

1.       In several places, some of the paragraphs have one or two sentences. It’s very distracting to the reader.

2.       The formula intake between pHF2.0 and pHF2.27 appears to be significantly different at 1 and 2 months of age (Supplemental Table 3).  Explain how this can affect growth?

3.       Any explanation on why the pH2.0 group did not compare to pH2.27 as expected (lines 336-337)?

4.       Discuss if changing the carbohydrate content impacts the growth in addition to the protein changes?  Is it possible that metabolism is affected in infants because of carbohydrate differences in formulas and what is the long-term impact in switching the source of calories to carbohydrates?

Author Response

Response to Reviewer 1 Comments

We want to thank the reviewers for their positive and constructive comments to our manuscript.

All of the reviewer’s comments were fully addressed by specific revisions of the manuscript. We are confident that these adjustments have improved the clarity and readability of the manuscript. We have formulated specific responses to each of the comments which hopefully provides an adequate level of feedback to any of the questions posed.

1.    In several places, some of the paragraphs have one or two sentences. It’s very distracting to the reader.

Response 1:

We reduced the number of paragraphs as suggested by the reviewer.

2.    The formula intake between pHF2.0 and pHF2.27 appears to be significantly different at 1 and 2 months of age (Supplemental Table 3). Explain how this can affect growth?

Response 2:

The formula intake in ml/day at 1 and 2 months is statistically significantly lower in pHF2.0 than in pHF2.27 (~60 ml/day).  When taking the body weight into account (ml/kg/day), the statistical difference disappears.

In term infants, formula intake is positively related to body weight, but growth velocity is mainly related to relative formula intake ml/kg/day. The small and statistically not significant difference in formula intake at 1 month of 169 versus 180 ml/kg/day account for 7 ml/kg/day could contribute partially to the lower weight gain in pHF2.0. By contrast, we also observed an identical intake in pHF1.8 of 169 ml/kg/day compared to HF2.27 without observing similar differences in growth.  After 1 month, the formula intakes (ml/kg/day) were very similar in the three groups.

During the statistical process, we applied an additional model (data not shown in the manuscript), which included formula intake as covariate, but the conclusions for the equivalence analyses did not change considerably and did not explain the differences between the study groups. We added this information to the manuscript.

3.    Any explanation on why the pH2.0 group did not compare to pH2.27 as expected (lines 336-337)?

Response 3:

We could not identify any explanation for the observed differences between pH2.0 and pH2.27. As indicated in the manuscript section 3.1 there are no apparent differences between the study groups when evaluating subjects and parental characteristics, neither between pH2.0 and pH2.27, nor between pH2.0 and pH1.8. Thus, there might be, as hypothesized in lines 390-393, factors we did not sufficiently controlled for, like unknown study group differences.

As described above, the differences in formula intake do not explain the difference in growth.

A potential explanation might be that the collected information on formula intake does reflect the reality. To reduce the burden for the parents, the formula intake was not collected daily but via four7-day diaries. Parents might not have reported the formula intake honestly and/or the seven days might not sufficiently reflect the actual intake.

As a part of the study design, but not described in this manuscript, bottles prepared by the mother/parents at their infants age of 1 and 4 months were analysed for their fat and protein content. These data confirm a similar fat and a different protein content between the study groups, they also showed some variability in the formula density. Such variability could play an additional role in the variability of the energy intake.

Another potential driver of the differences found is the low number of subjects in group pH2.0 (N=46), even though the number is close to estimated sample size during the interim analyses (48 subjects per arm, 69 per arm when compensating for drop out). However, we would expect that the low sample size would result in wider confidence intervals, it does not explain the identified differences in weight gain.

4.    Discuss if changing the carbohydrate content impacts the growth in addition to the protein changes? Is it possible that metabolism is affected in infants because of carbohydrate differences in formulas and what is the longterm impact in switching the source of calories to carbohydrates?

Response 4:

We acknowledge the fact that quality and quantity of carbohydrates in early life matters and can have long term effect on metabolic health.

All compositions of all three formulas is in line with the directive which requires a minimal lactose content of 4.5g/100kcal. In the three products, lactose is providing respectively 41, 42 and 43% of energy.

There is only a very small difference in carbohydrate (i.e. lactose) content between the three formulas with a maximum difference of 0.5g/100kcal (2%) in pHF1.8 compared to pHF2.27.

Such a minimal change has probably no direct or long-term impact on energy metabolism.

Our study suggests that a minimal change in protein energy ratio from 2.27 to 1.8g/100 kcal induces growth trajectories more similar to that of breastfed WHO reference with a small reduction in BUN.

References:

Srinivasan M, Mitrani P, Sadhanandan G, Dodds C, Shbeir-ElDika S, Thamotharan S, Ghanim H, Dandona P, Devaskar SU, Patel MS: A high-carbohydrate diet in the immediate postnatal life of rats induces adaptations predisposing to adult-onset obesity. J Endocrinol 2008, 197(3):565-574.

Vadlamudi S, Hiremagalur BK, Tao L, Kalhan SC, Kalaria RN, Kaung HL, Patel MS: Long-term effects on pancreatic function of feeding a HC formula to rats during the preweaning period. Am J Physiol 1993, 265(4 Pt 1):E565-571.

Reviewer 2 Report

This randomised, controlled, double blind equivalence trial compared the nutritional adequacy of a currently marketed partially hydrolysed whey-based formula containing 2.27 g/100 kcal protein with two intervention partially hydrolysed whey-based formulae with -10% (2.0 g/100 kcal) and -20% (1.8 g/100 kcal) protein content. It was concluded that all three formulas support adequate growth and are well tolerated up to 4 months of age and at 12 months of age.

Nutritional adequacy of partially hydrolysed whey-based formulae with low protein content (1.8 g/100 kcal) similar to formulae with intact protein is not consensual. Thus, the results from this study have interest, despite they are not novel since Ahrens et al. (ref. 25 of the manuscript) already have reported adequate growth in infants fed a partially hydrolysed whey-based formula with a low protein content of 1.9 g/100 kcal compared with a similar formula with 2.3 g/100 kcal.

Some aspects need to be addressed and clarified:

-  (line 83): Macrosomic infants, that is, birth weight >4,000 g have been included in the study. This may have introduced a bias since these infants may have had abnormally excessive intrauterine growth which may be an independent risk factor for postnatal overweight (Voerman 2019). This aspect should be adequately addressed in Discussion.

- (line 106): the statement “Infants were to be fed ad libitum and exclusively with the study formula…” is confusing. In which of the study formulae were the infants fed ad libitum? Which intake has been allowed to infants fed control formula? If ad libitum feeding has not been allowed equally in the three arms of the study this may have introduced a bias and should be acknowledge as a limitation.

- (line 140-142): The equivalence analysis was performed applying predefined margins of ±3 g/d mean weight gain. What was the rationale for these margins? Or it was a convenience criterion? Please clarify the definition criterion used in the manuscript.

- (line 322-331): It is stated that based on the reported 15 serious adverse events occurring in 7.2% of infants no safety concerns were raised for the tested formulae. The proportion of affected infants is not negligible and this may be a subjective judgment; therefore, the 15 adverse events should be specified.

- (lines 380-387): differences in total protein intake have been proportional to BUN, but did not drive growth outcomes proportionally, as the mean weight gain was lowest in pHF2.0, whereas in pHF1.8 it was intermediate. As potential explanations for these unexpected results, differences in study sample characteristics between groups and different reconstitution practices affecting the density of the formula and volume intakes were hypothesized. If the Authors hypothesize the presence of these factors, they are non-controlled factors of the trial and should be acknowledged as a limitation.

Minor critics

- (lines 356) Where it is stated “Alexander [27] and Beghin [28]” it should be “Alexander et al. [27] and Beghin et al. [28]”, therefore (line 360) replace “He” with “They”.

- (line 372) “postnatal” is redundant and should be eliminated

- (line 302) only “BUN” should be used since the abbreviation has been explained previously (line 134)

-  Tables 2 and 3 are out of order, since Table 2 is cited in the text (line 265) after  Table 3 (line 237)

Reference

-  Voerman E, Santos S, Patro Golab B, et al. Maternal body mass index, gestational weight gain, and the risk of overweight and obesity across childhood: An individual participant data meta-analysis. PLoS Med. 2019 Feb 11;16(2):e1002744.

Author Response

Response to Reviewer 2 Comments

We want to thank the reviewers for their positive and constructive comments to our manuscript.

All of the reviewer’s comments were fully addressed by specific revisions of the manuscript. We are confident that these adjustments have improved the clarity and readability of the manuscript. We have formulated specific responses to each of the comments which hopefully provides an adequate level of feedback to any of the questions posed.

1.(line 83): Macrosomic infants, that is, birth weight >4,000 g have been included in the study. This may have introduced a bias since these infants may have had abnormally excessive intrauterine growth which may be an independent risk factor for postnatal overweight (Voerman 2019). This aspect should be adequately addressed in Discussion.

Response 1:

Investigators were requested to enrol infants with a birth weight of ≥2.5 kg and ≤4.5 kg only when those infants were also considered as healthy. These criteria are commonly used inclusion criteria (see references 26, 29, 30, 32).

We confirm that 13 infants were included in the per protocol dataset (15 in total were randomised) with a birth weight of >4000 g, in pH1.8 three girls and four boys, in pH2.0 two boys and in pH2.27 two girls and two boys.

Considering the WHO Child growth standards, it can be expected that approximately 5% of girls and 10% of boys are born with a weight of >4000 g. In our dataset the girls with a birthweight of >4000 g represent 3%, the boys 5%. Therefore, we consider our dataset as the range of the WHO Child growth standards.

We acknowledge that birth weight may affect the growth trajectories. We investigated birth weight as a potential covariate in the analysis of the primary parameter, but it turned out to be non-significant. Similarly, we investigated the potential effect of pre-pregnancy weight of mother, smoking of mother during pregnancy and maternal diabetes, but similarly, there was no effect on the primary parameter. We added this information to the manuscript.

We assume that the intervention period of 4 months, the small sample size and the especially the observational period until 12 months might be too short to find a potential effect as suggested by Voerman.

2.(line 106): the statement “Infants were to be fed ad libitum and exclusively with the study formula…” is confusing. In which of the study formulae were the infants fed ad libitum? Which intake has been allowed to infants fed control formula? If ad libitum feeding has not been allowed equally in the three arms of the study this may have introduced a bias and should be acknowledge as a limitation.

Response 2:

In all the three groups recommendations to the mothers were to feed their infants ad libitum without any restriction.

We amended the sentence: “All infants were to be fed ad libitum and exclusively with the formula…” to avoid the assumption that only the test, but not the control formula was to be fed ad libitum. The feeding instructions were the same for all 3 formulas.

3.(line 140-142): The equivalence analysis was performed applying predefined margins of ±3 g/d mean weight gain. What was the rationale for these margins? Or it was a convenience criterion? Please clarify the definition criterion used in the manuscript.

Response 3:

The American Academy of Pediatrics Committee on Nutrition Task Force on clinical testing of infant formulas (AAP, 1988) concluded that “rate of gain in weight gain is the single most valuable component of the clinical evaluation of infant formula” and recommended that a weight gain difference greater than 3 g/day over 3 to 4 months should be considered nutritionally significant.

Based on this recommendation we set the equivalence margins for the primary outcome parameter at ±3 g/d.

We corrected the reference used.

4. (line 322-331): It is stated that based on the reported 15 serious adverse events occurring in 7.2% of infants no safety concerns were raised for the tested formulae. The proportion of affected infants is not negligible and this may be a subjective judgment; therefore, the 15 adverse events should be specified.

Response 4:

Information about the serious adverse events has been added to the text

5. (lines 380-387): differences in total protein intake have been proportional to BUN, but did not drive growth outcomes proportionally, as the mean weight gain was lowest in pHF2.0, whereas in pHF1.8 it was intermediate. As potential explanations for these unexpected results, differences in study sample characteristics between groups and different reconstitution practices affecting the density of the formula and volume intakes were hypothesized. If the Authors hypothesize the presence of these factors, they are non-controlled factors of the trial and should be acknowledged as a limitation.

Response 5:

We added to the manuscript that we acknowledge that some information might have been insufficiently captured via diaries.

6.(lines 356) Where it is stated “Alexander [27] and Beghin [28]” it should be “Alexander et al. [27] and Beghin et al. [28]”, therefore (line 360) replace “He” with “They”.

Response 6:

We revised the manuscript accordingly.

7.(line 372) “postnatal” is redundant and should be eliminated

Response 7:

We revised the manuscript accordingly.

8. (line 302) only “BUN” should be used since the abbreviation has been explained previously (line 134)

Response 8:

We revised the manuscript accordingly.

9. - Tables 2 and 3 are out of order, since Table 2 is cited in the text (line 265) after Table 3 (line 237)

Response 9:

We revised the manuscript accordingly.